# Auricular Osseointegrated Implant Treatment: Basic Technique and Application of Computer Technology

**Hiromasa Kawana [1,*], Shin Usuda [2], Seiji Asoda [3], Tsuyoshi Kaneko [4], Kaoru Ogawa [5], Tomoki Itamiya [6], Kei Fuchigami [1], Koudai Nagata [1], Ryoji Kitami [1], Katsuhiko Kimoto [7] and Michael Truppe [8]**

[1] Department of Oral and Maxillofacial Implantology, Kanagawa Dental University, Yokosuka 238-8580, Japan; fuchigami@kdu.ac.jp (K.F.); nagata@kdu.ac.jp (K.N.); kitami@kdu.ac.jp (R.K.)
[2] Department of Dentistry and Oral and Maxillofacial Surgery, Tachikawa Hospital, Tokyo 190-8531, Japan; usushin@keio.jp
[3] Division of Oral and Maxillofacial Surgery, Department of Dentistry and Oral Surgery, School of Medicine, Keio University, Tokyo 160-8582, Japan; asoda@keio.jp
[4] Division of Plastic Surgery, National Center for Child Health and Development, Tokyo 157-8535, Japan; kaneko-t@ncchd.go.jp
[5] Department of Otolaryngology, Head and Neck Surgery, School of Medicine, Keio University, Tokyo 160-8582, Japan; ogawak@keio.jp
[6] Division of Curriculum Development, School of Dentistry, Kanagawa Dental University, Yokosuka 238-8580, Japan; itamiya@kdu.ac.jp
[7] Division of Prosthodontics and Oral Implantology, Department of Oral Interdisciplinary Medicine, Graduate School of Dentistry, Kanagawa Dental University, Yokosuka, Japan 238-8580; k.kimoto@kdu.ac.jp
[8] 3D Implantat Navigation, Vienna 1080, Austria; truppe@mtruppe.net
[*] Correspondence: kawana@kdu.ac.jp; Tel.: +81-46-822-8880

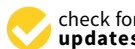

**Featured Application: Latest digital technology supported by anatomy and dental implantology is mandatory to accomplish safe and secure auricular osseointegrated implant treatment.**

**Abstract:** An epithesis using osseointegrated implants as an anchorage has been proven and established as an effective means for maxillofacial rehabilitation. In this paper, we describe the basic techniques of auricular epithesis and the applications of computer technology to its execution, ranging from diagnosis to surgery, and superstructure fabrication to maintenance. The key steps of this treatment are conducting Computed Tomography (CT) diagnosis before the operation to avoid implant penetration through the skull during the operation, embedding two implants into the section that represents the antihelix, and if possible, preserving or forming the tragus so that the edge of the epithesis is not close to the temporomandibular joint region to ensure mandibular movements are not restricted. We also discuss the applications of navigation surgery, which we are currently investigating, as well as the future prospect of augmented reality and mixed reality surgeries.

**Keywords:** epithesis; auricular epithesis; implant; osseointegrated implant; ear; navigation surgery; telenavigation surgery; intraoral scanner; digital impression; augmented reality; mixed reality

## 1. Introduction

Although auricular epithesis treatment is typically prescribed to patients with auricular deformities or defects who were unable to undergo plastic surgery during childhood, there have been several issues concerning its application. Some concerns are that the attachment of an external ear is required and the use of an undercut or the dependence on the adhesion provided by an adhesive is not only

unable to provide a solid anchorage to the epithesis, but the long-term use of adhesives can cause degradation of the epithesis and/or dermatitis, which has been a major problem for both patients and clinicians. Responding to these issues, Tjellström and Brånemark et al. applied osseointegrated titanium implants (hereinafter "implants") as the anchorage for the auricular epithesis in 1977, which resulted in a significantly improved durability [1]. Currently, it is unthinkable to attach an auricular epithesis without the use of implants. Since then, many articles have been published regarding auricular epithesis, but there are no reports of comprehensive treatment using digital technology jointly performed by maxillofacial surgeons, plastic surgeons, otolaryngologists, and anaplastologists. In this paper, we first outline standard treatments that are the basis of digital technology, and then discuss recent technologies such as navigation surgery and the future prospect of augmented reality and mixed reality surgeries (hereinafter AR/MR surgery), which we are currently investigating.

## 2. Methods

### 2.1. The Advantages and Disadvantages of Implant-Supported Auricular Epithesis Treatment

The advantages and disadvantages of implant-supported auricular epithesis treatment are easier to understand when compared to those of plastic and reconstructive surgery using autologous tissue. Some of the advantages of implant-supported auricular epithesis treatment are as follows: surgical invasion is minimal and it is possible to operate under local anesthesia depending on the case; the danger and probability of facial nerve damage are extremely low; it is unnecessary to collect autologous tissue; the number of operations can be clearly limited to two with just one implant placement and one implant fenestration; and the epithesis can be morphologically corrected, its color adjusted, and it can be refabricated non-invasively. In addition, patients are much more accepting of implant-supported auricular epithesis than of epithesis of the midface or the eye sections, as the ears are on the side of the face and are unimpacted by dynamic facial movements; furthermore, they do not play a role in facial expression creation. In contrast, some of the disadvantages of implant-supported auricular epithesis are as follows: the auricular epithesis is not part of the body, which necessitates the attachment and detachment of the epithesis; one needs to be meticulous in cleaning around the implant; there are risks of ambient inflammation around the implant and the epithesis may inadvertently fall off; and it is not suitable for pediatric patients as children are poor at cleaning and tend to dislike wearing artificial material.

### 2.2. Indications

The subjects for such a treatment are adults with cases of auricular deformity or defects who have experienced trauma, undergone a tumor resection, or had a congenital malformation, particularly those whom for some reason had no opportunity for auricular formation through costal cartilage grafting, artificial object insertion, or tissue expansion methods during their childhood.

### 2.3. Standard Treatment

#### 2.3.1. Collection of Materials

In preparation for treatment, one must consider a balanced location for the epithesis as part of the craniofacial organ, especially considering the symmetry of location and shape in relation to the unaffected ear when seen from the front. The basic materials that need to be gathered for this process are the facial photos (frontal, oblique, and lateral views) and a simple craniofacial CT scan, which are both mandatory. Furthermore, we recommend that one keep in mind simulation surgeries and model surgeries that may take place at a later time and either make sure to back up and store the CT data or, if the model fabrication occurs at an external facility, to have the data transferred to that location ahead of time.

### 2.3.2. Treatment Procedure

The following procedure assumes the auricular deformation or defect is on one side only:

Preoperative Diagnosis and the Operation Plan

In surgery planning, clinical anatomy-based selection of the location for implant placement is critical. First, during the facial observations, verify the distance between the outer canthus and the tragus (Figure 1), as well as whether the angle of the naso-auditory meatus line is symmetrical in relation to the bipupillar line. In most cases, they are symmetrical, and the location of the epithesis would be considered in such a way that the epithesis appears symmetrical in relation to the unaffected ear in both its placement and form. It is easiest to get a visual image by using mirroring technologies in stereolithography, by casting, or if an inverted model is created and prepared from the unaffected auricular based on the earlier wax-up (Figure 2). However, it is possible that the actual epithesis turns out to be asymmetrical in location, form, or color. It is important to sufficiently inform the patient and their family about the limitations of the treatment and obtain their consent. The implant site must be chosen so that perforation of the cavernous sinus and facial nerves does not occur (Figure 3); thus, the method of Tjellström et al. should be followed [2], as it considers the position of the epithesis based on its mechanical and aesthetic properties. In other words, this is a method that considers a safe implant site to be the site where the following is taken into consideration: the reference axis is set based on the straight line that connects the outer canthus and the tragus in the 3:00 to 9:00 direction of a clock, and a straight line that perpendicularly crosses the previous line centering the external auditory canal in the 0:00 to 6:00 direction also: if the right side is the affected side, the 7:30 to 8:30 and the 10:00 to 11:00 directions along with the location would form an antihelix. As the antihelix is located approximately 20 mm from the external auditory canal, during the operation, there will be a total of two implants: one implant in the 7:30 to 8:30 direction and the other in the 10:00 to 11:00 direction at a distance of approximately 20 mm away from the external auditory canal (Figure 4). While considering the above basics, the clinician would need to decide on the actual implant site based on clinical observations and findings from a CT scan. Next, the distance between the bone surface to the inside of the cranium is calculated using the CT images and models. Keep in mind that 4 to 8 mm implant bodies will be inserted from the temporal bone surface, the drill length is slightly longer than the implants themselves, and there is a chance that a technical error could cause the drill to go slightly deeper than intended. It would therefore be advisable to ensure the implant has at least 2 mm clearance to the inside of the cranium.

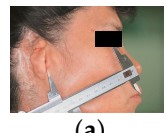 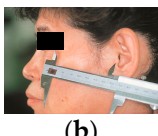

(**a**)  (**b**)

**Figure 1.** Distance between the external canthus and tragus on affected (**a**) and healthy (**b**) sides.

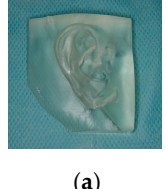 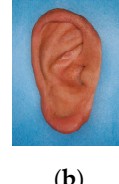

(**a**)  (**b**)

**Figure 2.** Mirroring stereolithographic model (**a**) and the epithesis of the same patient (**b**).

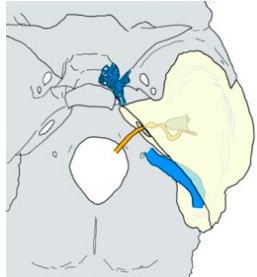

**Figure 3.** Anatomy of the cavernous sinus (netlike blue), the sigmoid sinus (tubular blue), the external auditory canal (conical gray), and the facial nerve canal (winding orange) in and around the temporal bone (pyramidal yellow).

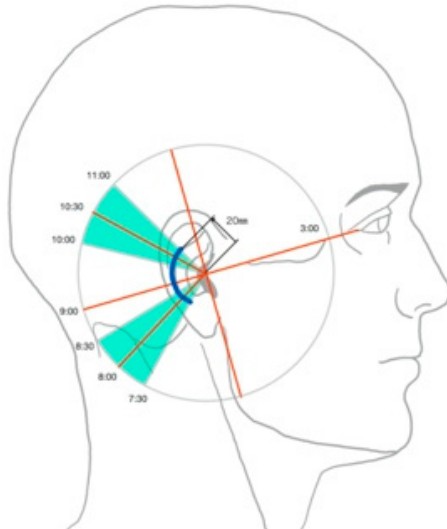

**Figure 4.** Safety area for implant placement described by Tjellström.

Implantation Surgery (Primary Surgery)

While it is possible to conduct the implantation surgery under general anesthesia, intravenous sedation, or local anesthesia, if one were to consider safety, general anesthesia would be the most ideal solution. An arc-like incision convexed towards the back is made approximately 25 mm behind the external auditory canal so that hair roots are not affected, and the periosteal flap is peeled off towards the front to expose the temporal bone surface where the antihelix is to be placed. Drilling occurs in the two spots identified on the basis of the previously mentioned measurements suggested by Tjellström. The drilling speed should be between 600 and 1200 rpm, and the process should be conducted while water is pumped to the drill site (Figure 5). After forming the initial pilot hole, a round-tipped probe should be used to check the bottom of the drill hole for bone-like hardness. Once it is verified that the hole does not extend to the inside of the cranium, the width of the hole could be increased. Once the final drilling is completed, the bottom of the hole should be checked with a probe once again. The implant bodies should be inserted using a low-speed engine at speeds between 20 and 30 rpm. Alternatively, a manual wrench could be used, and the torque should be set between 30 and 50 Newton cm (Ncm)(Figure 6). It is basic and standard practice to operate the engine at low speed and low torque while drilling or inserting implants, because these settings ensure that no heat damage to the bone occurs. For the same reason, one must not neglect the use of saline as cooling water while drilling. As for the implant systems, specialized maxillofacial systems can be found through Brånemark (Nobel Biocare AG, Kloten, Switzerland), Ankylos (Dentsply Sirona, Salzburg, Austria),

and Straumann (Straumann AG, Basel, Switzerland). In the case where a deformed auricular remains, appropriate measures should be taken to either keep, make morphological corrections, or resect so that what remains would be simple enough for the epithesis to be easily attached (Figure 7). In particular, the tragus should either remain or, if there is usable autologous tissue in the area, be created, as the tragus and anything in front of that area would not need to be covered in the future while using the epithesis. If the epithesis does not overlap with this section, the epithesis edges will not rub against the skin, as the skin over the temporomandibular joint would move with the opening and closing motion of the jaw. It should also be noted that it is mandatory to run another CT scan after the operation to check the location of the implant site and the depth of the tip of implant bodies (Figure 8).

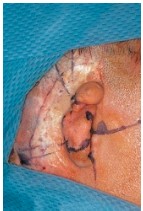

**Figure 5.** Preoperative marking according to Tjellström's standard.

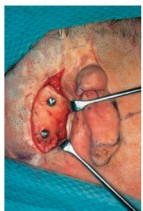

**Figure 6.** Implant placement.

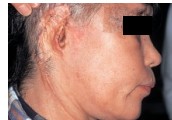

**Figure 7.** Postoperative situation. In this case, the deformed auricle was removed. The patient had also facial nerve palsy, and tapings around the eyelid were fixed after simultaneous eyelid suspension surgery.

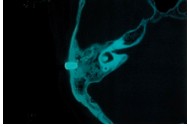

**Figure 8.** CT image after implant placement.

Implant Fenestration Surgery (Secondary Surgery)

The secondary surgery for implant fenestration is performed three to four months after the initial surgery to give enough time for osteosynthesis to occur. Most of these surgeries are conducted under local anesthesia. Incisions are made once again along the incision line from the implant surgery, excluding the periosteum, and the partial thickness flap is peeled back towards the front. Once the cover screw of the implants can be seen through the subperium membrane, a No. 12 scalpel is used to create an arch-like incision through the periosteum or a dermal punch is used to remove only the periosteum right above the cover screw while avoiding damage to the surrounding periosteum as best as possible. The cover screw is exposed, and the healing abutment is attached. It should be noted that a somewhat wider portion of the fat layer present around the abutment should be removed to

provide an environment comprised of attached fibrous tissue for the implants, which prevents ambient inflammation from occurring around the implants. Finally, when the partial thickness flap is returned to its original position while the abutment is attached, the skin directly above the abutment will be strained outwardly owing to the abutment, and a No. 11 scalpel should then be used to add a small incision at the same location and let the abutment penetrate the flap. The flap should be returned to its original location and sutured. If the small incision made for the abutment becomes accidentally torn, the area around the abutment should be sutured with a monofilament thread, such as nylon, in order to join the skin and the abutment. After this, the area around the exposed abutment, which is now penetrating through the skin, should be covered with an antibacterial ointment-impregnated gauze pressing down the tissue surrounding the abutment so that it will be fixed in place and adhere to the bone surface (Figure 9).

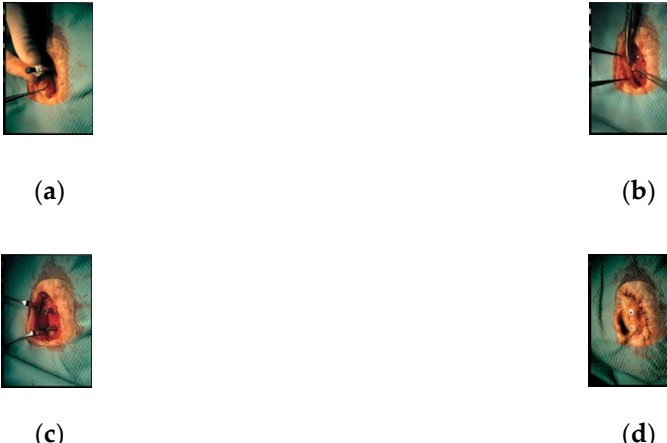

(**a**)                                                                                   (**b**)

(**c**)                                                                                   (**d**)

**Figure 9.** Second-stage fenestration surgery. (**a**) Punch out the periosteum just above the implants after raising a partial-thickness skin flap. (**b**) Excise surrounding adipose tissue. (**c**) Connect healing abutment. (**d**) After the flap closure, wrap gauze containing an antibacterial drug ointment in a figure of eight and apply pressure around two abutments.

Fabrication of the Auricular Epithesis

The fabrication of the epithesis begins one or two months after the secondary surgery once the skin has stabilized. After protecting the hair using a disposable cap and tape, the impression of the site is collected using a silicone-based dental impression material for implants (Figure 10) to create a plaster cast. Next, the wax model fabricated over the plaster model is checked for suitability at the chairside, and a color-matching test is conducted at the same time (Figure 11). Although it is necessary to fabricate the connector to the implant that supports the epithesis, we have adopted a design that adds a cantilever to the metal bar frame so the attaching and detaching can be more convenient and prevents subsiding around the edges of the epithesis. For this reason, we prefer to use a bar attachment that can connect the two implants together with metal (Figure 12). Additionally, as the metal frame is fabricated, it comes right below the antihelix, where there is some thickness to the epithesis to prevent the frame from being visible. The final adjustment of the edges of the epithesis is done directly after fitting, checking its adhesion and breathability.

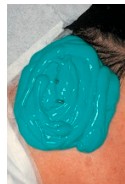

**Figure 10.** Taking impression with silicon impression material.

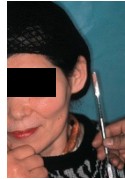

**Figure 11.** Color match test. Compare with the color on the healthy side.

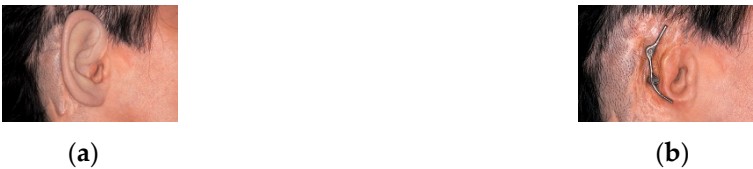

(**a**)　　　　　　　　　　　　　　　　　　　　　　　　　　(**b**)

**Figure 12.** The pinna epithesis (**a**) maintained by the bar attachment (**b**).

Follow-Up

Patients should be called back into the office for a follow-up appointment at least once every six months to check whether there are any signs of inflammation in the skin surrounding the penetrated abutment, whether there is slack between the epithesis and the metal frame, and that good hygiene has been maintained. When there are signs of ambient inflammation around the implants, it is likely hygiene has been unsatisfactory; if so, the patient should be instructed on how to correctly brush their implants and surrounding soft tissue. In the case of drainage, oral antimicrobial medicine should be prescribed, a pocket wash with saline or a benzalkonium chloride solution should be provided, and a follow-up visit after one a week should be requested for reexamination. After inflammation has been treated, brushing should resume; if a discharge continues to form, the antibacterial medicine should be changed and the patient should be monitored further. The localized application of antibacterial ointment is no more effective than disinfectant alone [3].

*2.4. Application of Computer Technology*

2.4.1. Navigation Surgery

Even with the above anatomical knowledge and careful preoperative preparation based on diagnostic imaging, errors could occur during the operation. To prevent such errors, a navigation system that allows one to capture the drilling directions and the depth of the tip of the drill in real time is desirable. There are several important points to note regarding navigation.

First, the CT data are obtained for use in the imaging anatomical diagnosis and simulation of the drilling positions (Figure 13). With regards to the preoperative simulation, after deciding on the location of the implant site based on the anatomical guidelines in Tjellström et al., one can use the CT images to determine if there is enough thickness to the bone to insert an implant. When planning for a navigation operation, in addition to the above, you will use the computer to create a mirror image of the unaffected ear and place this image on the affected side to get a sense of what the post-operative image may look like. For example, in the case shown in Figure 12, as a decision was already made to use the remaining elongated and deformed ear to surgically create a tragus; the mirroring of the unaffected ear helped select the section of the deformed ear that should be used as the tragus and the section that should be resected (Figure 14). Second, the surgical template with markers is fabricated; a resinous apparatus embedding two implant position markers at the drilling sites and twenty 0.5 mm metallic registration markers on the bar between the ear template and the mouthpiece fitting on the maxillary arch (Figure 15). The CT is taken again with the apparatus attached with three reflective spheres, and the registration is provided on the display (Figure 16). After the confirmation of accurate registration, the ear part of the apparatus is cut before surgery in order to let the implant positions be visible (Figure 17).

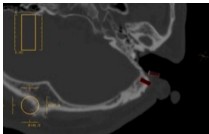

**Figure 13.** Preoperative implant placement simulation.

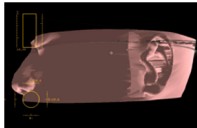

**Figure 14.** Using CT software, a mirror image of the healthy ear is placed on the affected ear, and the auricle resection site should be examined preoperatively.

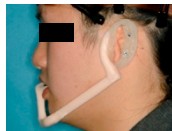

**Figure 15.** Diagnostic template. Two implant position markers are situated in the plate at the antihelix, and the mouthpiece is used as a stabilizer of the template.

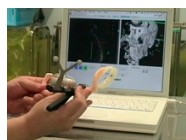

**Figure 16.** Preoperative registration. Since the marker is set in the template, preoperative registration can be performed without a patient.

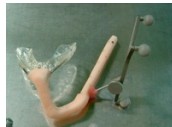

**Figure 17.** After reading the data on CT, the auricle portion of the template is separated by the time of surgery. This 21-year-old man had been suffering from left microtia and missed the opportunity of auricle plasty in his childhood due to severe pediatric asthma and had hidden his deformed auricle with long hair. Because he had to cut his hair short for an employment test, he wanted this treatment.

There are two main types of navigation systems: optical and magnetic-field type systems. We used the optical navigation system for the sake of convenience as it was pre-loaded in the dental implant software (Artma, Austria) (Figures 18 and 19). However, in a hospital setting, it is possible to use the navigation systems already used by the Department of Neurosurgery or the Department of Otolaryngology or apply the navigation system specifically designed for oral implantations if required. The advantages of using a navigation system along with conventional practice are as follows: the surgeon can verify during the operation, in real time, the drilling direction and its depth within the bone, which is impossible under conventional conditions, and thus avoid damaging vital organs (Figure 20). When the preoperative analysis and the reality differ during the operations, it is possible to reanalyze the images during the operation and change the implantation plan accordingly. By creating a surgical template attached to the maxillary dentition, one can start the preoperative registration without the presence of the patient, which reduces the operation time (Figures 16 and 17). The disadvantage is that there is a slight error margin between the CT and the location information detection system (which is said to be +0.35 mm with the CT slice width with the Artma system, though its competitor systems have a similar range of errors), which means one must gain proficiency in registration.

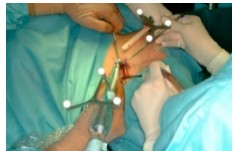

**Figure 18.** Navigation surgery.

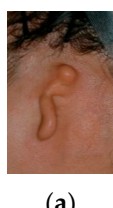　　　　　　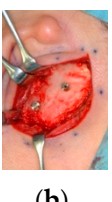　　　　　　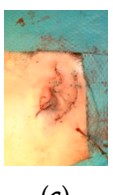

(**a**)　　　　　　　　　　　(**b**)　　　　　　　　　　　(**c**)

**Figure 19.** (**a**) Pre-, (**b**) intra-, and (**c**) post-operative findings. Two implants were placed and the tragus formed using autologous tissue.

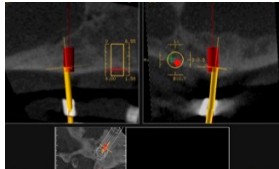

**Figure 20.** Navigation during drilling into the temporal bone. The drill (yellow bar) is guided to the expected position and direction (red rod).

### 2.4.2. Telenavigation Surgery

Telenavigation surgery is a type of navigation surgery that enables the induction of instrument directions and positions displayed on CT images sharing diagnostic and surgical information with very experienced specialists and engineers in remote locations. The preoperative plan is set up on in a teleconference via Internet communication with them. They also are on standby during surgery for providing any surgical support needed, as well as support with the software configuration in the case of a modification of the implant placement or any system trouble. They can carry out software operations via remote control, even if the surgical operator is not proficient in software operation (Figure 21).

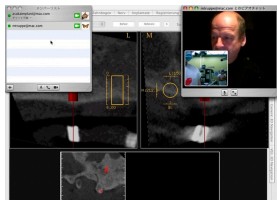

**Figure 21.** Telenavigation surgery. An experienced specialist is watching both the surgical field and navigation display on CT during surgery from Austria.

### 2.4.3. Model Surgery

Since CT scans are not only capable of constructing 3D images but also creating 3D models, 3D model allows for a preoperative 3D model surgery based on the patient's own skull. One can virtually perform the same operation under the same conditions by inserting dummy implants into the 3D model and verifying whether the drilling location, direction, and depths are appropriate (Figure 22). In addition, such a model surgery could be used to educate inexperienced clinicians. Additionally, the clinician may also bring the model into the operating room and verify its final location. As a

side note, considering the fabrication process of the epithesis, using the mirroring function to create a mirrored model of the unaffected ear significantly reduces the workload for the anaplastologists (Figure 23); furthermore, the final auricular epithesis model would be more realistic (Figure 24).

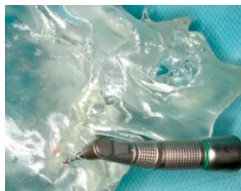

**Figure 22.** Preoperative model surgery using a stereolithographic model of the patient.

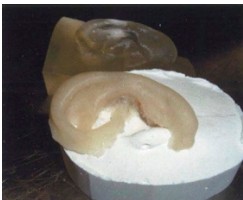

**Figure 23.** Mirrored model of the pinna on the healthy side (back) and its wax model (front).

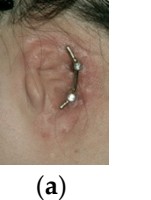

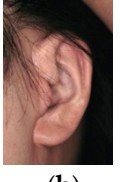

(**a**)                                     (**b**)

**Figure 24.** Completed (**a**) bar attachment and (**b**) auricular epithesis.

2.4.4. Optical Impression Acquisition

Instead of using conventional silicone-based dental impression materials to obtain impressions (Figure 10), optical impressions will be obtained using a camera. This eliminates any worry about leaving residue on the patient's face or hair from the impression materials and prevents the deformation of the model, which can be caused by distortions on the impression material. We utilized intraoral scanners that are currently used in dental treatments to obtain an optical impression, which was converted into STL data and used to create digital wax-ups (Figure 25). The drawback of this approach is that, if the camera touches the skin, the image becomes distorted; furthermore, some camera models are unable to scan into the deeper areas of the external auditory canal, which then remain unscanned (Figure 26). However, as the deeper inner structure of the ears is simple, this system limitation does not hinder fabrication of the epithesis.

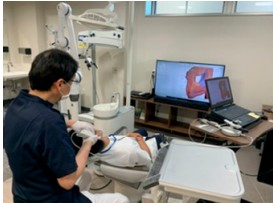

**Figure 25.** Optical impression by an intraoral scanner for dental use.

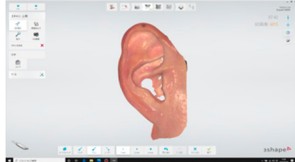

**Figure 26.** Image scanned by intraoral scanner. The shape and color are clearly reproduced.

2.4.5. AR/MR Surgery

In addition to the previously mentioned disadvantages of the navigation surgery, the operative field projected on the monitor tends to be flat and lacks three-dimensionality. As a result, when the monitor's location or its angles change, the images on the monitor might become distorted. Furthermore, the registration process is long, and the equipment, which can cost 30 million yen (approximately 280,000 USD), with an annual maintenance cost of a million yen (approximately 9300 USD), is prohibitively expensive. For this reason, only well-funded hospitals can introduce such equipment. Hence, to resolve these issues, wearable computer glasses for AR/MR operation have been developed*. However, the initial product was unsatisfactory owing to its poor hardware performance; for example, attempts to draw a heavy medical 3DCG model were unsuccessful because the framerate dropped. In addition, to display a 3D model, it was necessary to set a marker, and even when this was done, the location did not remain fixed. Furthermore, the initial product could only be controlled by a controller; thus, a surgeon would not be able to operate it. In contrast, although the viewing angle itself tended to be narrow, the HoloLens (Microsoft, WA, USA), which was developed later, has a high pixel density that resulted in clear images, and a corresponding surgical support system OpenSight (Novarad, UT, USA) was also developed (Figures 27 and 28). As long as STL data are available, several 3D model viewers can easily be made; furthermore, as long as the segmentations are performed, the 3D models themselves can be fabricated within a short time using Unity (Unity Technologies, California, CA, USA), allowing for the reduction of major expenses related to navigation systems and stereolithography prototyping systems.

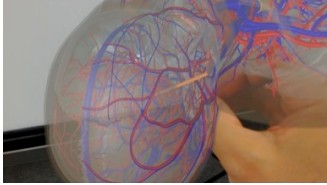

**Figure 27.** MR surgery. The alignment between the actual drill and the drill in the MR space (orange rod) is superimposed. UI warning display by contact judgment is also implemented.

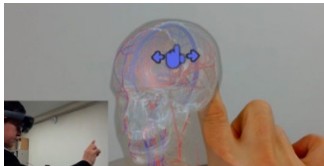

**Figure 28.** Medical 3D model viewer. Because the gesture recognition is equipped, the operator can rotate and move the model. This mechanism is effective for medical and dental education and surgical consideration.

However, as the gesture recognition accuracy is extremely delicate, it would henceforth be necessary to devise a better procedure for model placement. Furthermore, when aligning the 3DCG model in the HoloLens to the operative field, in the case of manual operation, gesture controls, an external keyboard, or a controller must be used. If the automatic alignment feature is used, a Vuforia (PTC, MA, USA) without a marker should be used to simply recognize the shapes; otherwise,

two- or three-dimensional markers should be used. However, the disadvantage is that surgical lighting is far too bright for the HoloLens so that it causes a halation, which reduces its recognition rate. To increase its recognition rate, it may be possible to consider bead-like marker templates to capture the feature points; however, in such a case, at least 400 marker points would be needed, which makes the fabrication itself difficult. For this reason, other methods were considered to resolve this issue, such as adding zebra-striped patterns on the templates or syncing with STL data of the above-mentioned intraoral scanner. However, as the STL data recognition function of the HoloLens is unstable, the solutions for alignment have been moving towards pasting different QR codes onto three-dimensional objects printed on a 3D printer. Even then, unless you are able to align the actual surgical instruments to the 3DCG instruments seen though the HoloLens, the proposed approach is not necessarily an easy solution; thus, several issues remain. Among them is the question of the examination and assurance of its accuracy, including what methods could be used to evaluate such accuracy. In addition to these issues of accuracy, the surgical use of the HoloLens they must undergo ethics reviews and obtain medical device certification as well. For these reasons, the application of the HoloLens in actual surgeries is limited. Nonetheless, some educational systems have already started to use the HoloLens for training purposes by simulating endonasal transsphenoidal surgery for treating pituitary tumors*3). There are several advantages to this, as such training can be conducted remotely, and the state and audio of the operation site can be shared using general TV conference call systems, among other options. Instructions and images data can be provided from remote areas as well, which allows for more detailed and solid communications. The disadvantage of the HoloLens is that you probably will not be able to wear it for more than an hour during the operation owing to simulator sickness and its weight. Moreover, as mentioned earlier, under surgical lights, the 3D display becomes less vivid and flattens the images. Furthermore, it has poor alignment accuracy, and for each user, the interpupillary distance (IPD) needs to be re-measured. Lastly, one needs to become proficient in gesture controls to use it. In the future, it will be necessary to consider further application developments, work in conjunction with the Picture Archiving and Communication Systems (PACS), and automate the applications, synchronizing them with conventional surgical simulation systems, and evaluating the safety and efficacy as a surgical system.

## 3. Discussion

In applying osseointegrated implants to auricular epithesis, it is important to ensure the future superstructure will look natural. This concept is a so-called top-down treatment, in which the treatment goal is set first and then the treatment strategy is determined to suit this goal. In doing so, one must select the location of the implant insertion so that the future superstructure becomes functional and is aesthetically pleasing; furthermore, the implant must be set in such a way that the patient can easily maintain good hygiene. If one were to take a bottom-up approach in treatment, such as selecting the insertion site of the implants without considering the future position of the superstructure and then somehow fabricating an epithesis, one would soon find that the support structure for the epithesis will become more complicated. Moreover, the form of the epithesis itself would lose its natural look. Furthermore, the patient may not be able to properly maintain the site by conducting simple brushing, which may cause ambient inflammation around the implant, which, if left untreated, leads to chronic inflammation and, in the worst case, may even lead to the implant itself to falling out. While such occurrences could be said to be the drawbacks of implants that penetrate the skin, errors must be avoided in devising the surgical plan and designing the superstructure. Although it has been reported that the application of osseointegrated implants to auricular epithesis had a high survival rate of 99.4% out of the 355 Brånemark implants inserted over eight years [2], it has also been found that the potential implant site on the temporal bone is extremely narrow. Thus, even the slightest of drilling errors could lead to the inability of the implant bodies to take hold, never gaining the initial stability. As a result, fibrous tissue could interpose itself in the slight gap between the implant body and the bone; osseointegration of the implants would not occur, which then may lead to the implant falling

out [4]. This would be an issue based on a technical error, which also occurs in standard dental implant procedures. At the same time, the inexperience of the clinician is a risk factor said to invite failures [5]; and this inexperience increases the failure rate of the implant before it even bears a load [6]. Even when an inexperienced surgeon performs the procedure under the guidance of an experienced practitioner, the failure rate for the implants is not still high [7], indicating that clinical experience alone cannot be the deciding factor in successful implantation. However, one cannot deny that implant surgeries involving auricular epithesis are not common in many medical facilities. In the case of our medical group, although our oral and maxillofacial surgeons performing implant insertions have participated in over 6000 oral implantations, when dealing with the formation of the residual auricular or in the field of anatomical knowledge concerning the temporal region, plastic surgeon, otolaryngologist and oral and maxillofacial surgeon generally collaborate. In addition, we consult with overseas facilities as their surgeons are much more experienced with the navigation system. In short, we first take safety into consideration even as we take an interdisciplinary approach in the treatments we provide.

In patients with auricular deformities or defects, it is common to see malformations in the inner and outer ears. For this reason, it is important to consult the preoperative CT scan, which can assist clinicians in understanding the structure deep inside the ear; thus, collaboration with an ENT specialist becomes vital. Additionally, thickness figures of the temporal bone obtained using the CT measurement software can be used to make sure that the implants have the correct lengths, which will prevent the implant from perforating the inner cranium [8]. Furthermore, although Tjellström et al. recommended the insertion of three to four implants, a limited number of supports for the implants can bear the weight of the epithesis; moreover, as cleaning becomes difficult when too many implants are inserted in a narrow area, two implants are appropriate, and the current recommendation is that each implant body be more than 15 mm apart from the next one [9]. In secondary fenestration surgery, it is important to remove the fat surrounding the implants. According to Tjellström et al., fat must be removed during the implant insertion operation. However, we attempted to remove the fat during the secondary operation after the osseointegration was complete to prevent the implant body from penetrating the skin above it, which could lead to its exposure and infection [1].

With regard to the application of the navigation system, great progress has been made so that safer operations can now be performed because the depth of the tip of the drill can be visualized. Familiarity with the preoperative preparations, the margin of error in the positioning during operation, and responses to the stoppage of the operation remain issues that require further consideration.

It is evident that operation simulations on a flat screen using CT cannot consistently reflect realistic operation procedures. By performing a model surgery in which a dummy implant is placed by drilling deep into the model skull while looking through it, a more realistic surgery simulation can be performed [10]. As this simulation surgery can be performed by the assistant, it would also fulfill an educational purpose. Going forward, we wish to reduce or maintain the costs of surgery, and we would like to be engaged in the development of AR/MR technologies [11], haptic technologies [12], and scanning technologies that are not dependent on the skill of the surgeon and anaplastologist, in addition to educational simulators.

**Author Contributions:** Conceptualization, H.K.; Methodology, H.K.; Software, T.I. and M.T.; Validation, H.K., S.U., S.A., T.K., K.O. T.I., K.F., K.N., R.K. and T.M.; Formal Analysis, H.K., S.U. and T.I.; Investigation, H.K. and S.U.; Resources, H.K. and T.K.; Data Curation, H.K., S.U. and T.I.; Writing—Original Draft Preparation, H.K.; Writing—Review & Editing, H.K.; Visualization, U.S., T.I. and K.K.; Supervision, T.K. and K.O.; Project Administration, H.K.; Funding Acquisition, H.K. All authors have read and agreed to the published version of the manuscript.

**Funding:** This communication received no external funding.

**Acknowledgments:** The authors are grateful to Tetsuo Takizawa, CEO of Meditech and anaplastologist, for making the epitheses. The authors acknowledge the contributions of Katsuhiko Yamaya, Toshifumi Nakashizu, and Kazuya Suzuki, dental technicians of Kanagawa Dental University Hospital, for their technical assistance in intraoral scanning. The authors thank the publisher, ZEN·NIHONBASHI·SHUPPANKAI, for allowing us to use some of the figures published in a Japanese magazine, PEPARS No. 42, that the corresponding author wrote in 2010. The authors would like to thank Editage (www.editage.jp) for the English language review.

**Conflicts of Interest:** The authors declare no conflict of interest.

**Ethical Statement:** According to the 'Personal Information Protection Low' in Japan, all patients gave their informed consent for being exhibited in the article and the review was conducted in accordance with the Declaration of Helsinki.

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
