# Peer review of "Auricular Osseointegrated Implant Treatment: Basic Technique and Application of Computer Technology"

_applsci, doi:10.3390/app10144922_

Round 1
Reviewer 1 Report
Dear authors,
your paper treats an interesting subject. However, different changes should be done. First of all, the english language should be revised by a native english speaker as different mistakes are present on the whole text.
Introduction:
please be careful when you cite a paper. Line 54 you reported that Brånemark et al. (1979) ...meanwhile the reference is reported as 1. Tjellström, A.; Jansson, K.; Brånemark, P-I. Craniofacial Defects. Application in the maxillofacial 409 region. Advanced osseointegrated surgery.; Worthington, P.;Brånemark, P-I. Eds; Quintessence: Chicago, USA, 1992; pp.293-312. Please carefully revise all the references. Then please state why it is necessary this paper to the research worldwide and describe the state of the art.it is not reported in this section.
Moreover, the paper seams more likely to be a communication than a review. In a narrative review the authors should make the point of the subject and treat it through what it is reported in the literature. Here seams that only author considerations are reported and very few studies.No other studies, no online or hand search has been performed. It is really difficult for the reader to understand what the authors have done in this review. Please report also what the worldwide literature says for each of the sections you mentioned in the paper, otherwise change the type of your paper from review to communication.
Reviewer 2 Report
It is an interesting topic and a good comprehensive review about the epithesis of ear using upgraded current technologies; implant, navigation surgery, and augmented and mixed reality surgeries. However, the writing format looks not clear as a review paper, please clarify the writing format as a review paper if it is. In addition, it would be better off more focusing on the preparation of treatment planning using digital technologies and navigation sugery than conventional epitheis of ear with implants. the AR/MR section should be more thoroughly reviewed if it is thought to be necessary to be included.
Please provide the IRB approval or exemption, if you want to include the clinical cases and patients pictures.
It would be better to write up this manuscript more neatly and consisely focusing on the current and future techniques for epithesis treatments of face.
Round 2
Reviewer 1 Report
The authors have adequately addressed the concerns of this reviewer.
Reviewer 2 Report
If you want to change the format of this manuscript from a review to a communication, it would be better to write this manuscript more concisely.
Please checks typos and grammar.